# Host Factor Rab4b Promotes *Japanese Encephalitis Virus* Replication

**DOI:** 10.3390/microorganisms12091804

**Published:** 2024-08-31

**Authors:** Qin Zhao, Chang Miao, Yi-Ting Chen, Long-Yue Zhu, Ya-Ting Zhang, Sai-Qi Luo, Yu-Luo Wang, Zhu-Ming Zhu, Xinfeng Han, Yiping Wen, Rui Wu, Senyan Du, Qi-Gui Yan, Xiaobo Huang, Shan Zhao, Yi-Fei Lang, Yiping Wang, Yi Zheng, Fei Zhao, San-Jie Cao

**Affiliations:** 1Research Center for Swine Diseases, College of Veterinary Medicine, Sichuan Agricultural University, Chengdu 611130, China; 2National Demonstration Center for Experimental Animal Education, Sichuan Agricultural University, Chengdu 611130, China; 3Sichuan Science-Observation Experimental Station of Veterinary Drugs and Veterinary Diagnostic Technique, Ministry of Agriculture and Rural Affairs, Chengdu 611130, China

**Keywords:** Rab4b, *Japanese encephalitis virus*, host factor, viral replication

## Abstract

Although the *Japanese encephalitis virus* (JEV) infects various cell types, its receptor molecules are still not clearly understood. In our laboratory’s prior research, Rab4b was identified as a potential host factor that facilitates JEV infection in PK15 cells, utilizing a genome-wide CRISPR/Cas9 knockout library (PK-15-GeCKO). To further explore the effect of Rab4b on JEV replication, we used the *Rab4b* knockout PK15 cell line using the CRISPR/Cas9 technology and overexpressing the *Rab4b* PK15 cell line, with IFA, RT–qPCR, and Western blot to study the effect of Rab4b on viral replication in the whole life cycle of the JEV. The results show that the knockout of *Rab4b* inhibited the replication of the JEV in PK15 cells, and the overexpression of *Rab4b* promoted the replication of the JEV in PK15 cell lines. Furthermore, we demonstrated for the first time that host factor Rab4b facilitates the adsorption, internalization, assembly, and release of the JEV, thereby promoting JEV replication. This study enriches the regulatory network between the JEV and host factors and lays the experimental foundation for further understanding of the function of the Rab4b protein.

## 1. Introduction

The *Japanese encephalitis virus* (JEV) causes *Japanese encephalitis* (JE), an acute zoonotic disease [1]. In domestic animals, including humans, the JEV is a vector-borne zoonotic virus that causes encephalitis [2]. In humans, the primary clinical manifestations and sequelae of JE predominantly encompass fever, convulsions, and alterations in sensory organ function [3]. Among those who develop the disease, approximately one-third recover completely, one-third experience severe lifelong neurological complications, and one-third succumb to the illness. Patients presenting with meningoencephalitis may progress to permanent neurological deficits or ultimately die [4]. Additionally, patients may present with rare conditions, including flaccid paralysis [5]. About 20% to 40% of JE patients die during the acute stage, and 50% are left with severe neurological sequelae [6]. Patients typically have cognitive impairment, which is a major problem affecting quality of life [7]. Pigs serve as crucial amplification and reservoir hosts for the JEV, with infections in pigs predominantly being subclinical [1]. The repercussions of JEV infection are notably significant in swine populations. In sows, JEV infection can lead to abortion, mummified fetuses, and stillbirth, while in boars, it can result in orchitis and a decline in semen quality, collectively causing substantial economic losses to the pig industry [8]. Although adult swine typically do not exhibit symptomatic disease following infection, the JEV remains a major reproductive concern, leading to abortion, stillbirth, and congenital anomalies. Furthermore, infected piglets may present with fatal neurological disease [9]. To prevent JE, immunoprophylaxis is considered the most effective method.

The JEV belongs to the family *Flaviviridae* and is a single-stranded RNA virus with a positive sense [10]. The JEV is mainly transmitted by *Culex tritaeniorhynchus* in regions where the virus is endemic, but other mosquito species, especially within the *genus Culex*, can act as vectors [11]. In addition, field surveys and laboratory assessments indicate that *Ar. subalbatus* is a major vector of JEV transmission [12]. Typically, human infection is caused by the Culex species infected by the JEV, transmitted in enzootic cycles between mosquitoes, pigs, and birds [13]. JEV infection is a primary global public health concern [14]. Due to the significant detrimental impact of the JEV on both animal populations and public health, strengthening JEV research remains a top priority for animal disease prevention and control and for ensuring human health. Viruses highly depend on the host’s cellular metabolism to support their replication [15]. For JEV replication, the host cells’ metabolism plays a critical role. Two factors determine the severity and clinical outcome of JEV infection: viral and host factors. For the JEV invasive body, different cell types, attachment factors, and receptors have been identified. Several molecules have been proposed as cell receptors to the JEV, including Heparan sulfate glycosaminoglycans (HSPG), HSP70, and HSP90B, vimentin, CD4, laminin receptor, and α5β3 integrin [16]. Further research disclosed that Glycosaminoglycans (GAGs) / glucose-regulated protein 78 (GRP78) / HSPG act as attachment factors for the JEV in BHK-21 cells, C6/36 mosquito cells, CHO-K1 (derived from Chinese hamster ovary) cells, mouse neuronal (Neuro2a) cells, mouse primary neurons, and human epithelial Huh-7 cells [17,18]. HSP90β was demonstrated as a receptor for the JEV in Vero cells [19]. PK15 cells’ and BHK-21 cells’ surface vimentin has proven to be the receptor for JEV entry and interacts with the E protein of the JEV [20]. Low-density lipoprotein receptor (LDLR) is a possible cellular receptor for the JEV since it is involved in JEV entry into the A549 cells and can bind to JEV-E [21]. The fraktalkine receptor CX3CR1 was up-regulated in human microglia following exposure to both the JEV vaccine and live JEV [22]. RIPK3 promotes JEV replication in neurons by downregulating IFI44L [23]. Although previous studies have identified the target cells and receptors for JEV infection, the detailed molecular mechanism for JEV entry into host cells remains unclear [24,25]. Therefore, this issue requires further investigation.

Rab4b, an evolutionary conserved GTP binding protein, contains the Ras domain [26]. As an early endosomal Rab, Rab4b localizes to sorting endosomes and mediates “fast” endosomal recycling [27]. Previous research showed that *Rab4b* genes played roles in host defense against pathogen infection, such as Rab4b has been implicated in the processes of virion assembly and the release of infection in human cytomegalovirus [28], the *Rab4b* gene is upregulated in channel catfish infected with *Edwardsiella ictaluri* [29]. When *Chlamydia pneumoniae* enters host cells via EGFR-dependent endocytosis into an early endosome, immediately after entry, the early chlamydial inclusion acquires early endosomal Rab4b [30], and *Rab4b* genes involved in vesicle and endosomal transport in JEV infection in human brain microvessel endothelial cells [31]. JEV amplification occurs in pigs, and most animals are asymptomatic until they become infected [1]. Previous studies show that host factors are crucial to JEV replication [23]. However, research into the swine host factor Rab4b, which promotes JEV infections, is lagging. PK15 is a cell line exhibiting epithelial morphology that was isolated from the kidney of an adult pig, and PK15 cells are a widely utilized model for the propagation of the JEV [32,33,34,35,36]. It is possible to identify host genes involved in JEV infection using pig kidney cells (PK15 cells) [36,37]. 

Earlier work identified Rab4b as a host factor related to JEV replication based on CRISPR library screening. In this study, we utilized the PK15 cell line, the *Rab4b* knockout PK15 cell line, and the *Rab4b* overexpressing PK15 cell line. We employed CCK-8 assays, immunofluorescence assays (IFAs), reverse transcription–quantitative polymerase chain reaction (RT–qPCR), and Western blot analysis to investigate the impact of Rab4b on viral replication throughout the life cycle of the JEV. Our objective was to elucidate the roles that Rab4b plays in this context and to understand its function and role further. Rab4b is responsible for promoting the replication of the JEV, and this study deepens our understanding of this mechanism.

## 2. Materials and Methods

### 2.1. Cells, Virus, and Plasmid 

Porcine kidney PK15 cell lines (ATCC no. CCL-10, Manassas, VA, USA), the *Rab4b* gene knockout PK15 cell lines (PK15-Rab4b-K.O.) by using CRISPR/Cas-9 gene-editing technology, and the PK15 cell lines with stable overexpression of Rab4b (PK15-Rab4b-O.E.) were made and kept in our laboratory [38]. All cells were cultured in Dulbecco’s modified Eagle’s medium (DMEM; Gibco Grand Island, NY, USA) and supplemented with 10% fetal bovine serum (FBS; Gibco, Grand Island, NY, USA) containing 100 U/mL penicillin, 100 μg/mL streptomycin (Gibco, Grand Island, NY, USA) with 5% CO_2_ at 37 °C. 

The genotype III JEV strain (SA14) was used and maintained in our laboratory, and the viral titer was 5.0 × 10^7.0^ PFU/mL, GenBank accession U14163.1.

The full-length infectious cDNA clone plasmid of JEV pACYC-JEV-SA14/U14163 was a kind gift from Prof. Bo Zhang (Wuhan Institute of Virology, Chinese Academy of Sciences, Wuhan, China), and the construction of the JEV pACYC-JEV-SA14/U14163 infectious cDNA clone plasmid was performed as described previously [39]. 

The plasmid for the expression of Rab4b pCDNA3.1-flag-Rab4b was made and kept in our laboratory [38].

### 2.2. An Assay for Determining Cell Viability Using the CCK-8 Kit

An assay for cell viability was conducted using CCK-8 (Beyotime Biotechnology, Shanghai, China). Seeded PK15-WT, PK15-Rab4b-K.O., and PK15-Rab4b-O.E. cells were inoculated with JEV (MOI = 0.1) in 96-well plates. After 12 h and 24 h of culture, to each well, 10 μL of CCK-8 reagent was added and the whole was incubated for one hour at 37 °C. With a microplate reader at 450 nm, each well’s absorbance was measured. 

### 2.3. Real-Time–Quantitative PCR (RT‒qPCR) Assay for Virus Replication

JEV is an RNA virus transmitted by mosquitoes that invades target cells through its envelope protein, JEV-E [40]. The E protein is encoded by the JEV E gene [41]. In the JEV and other yellow viruses, the E protein is the most important structural protein. Through fusion with the cell membrane and interactions with receptors on the membrane, the virus can enter the cell and be neutralized by antibodies [42,43]. Thus, a high level of E gene expression can provide a good indicator of JEV replication and infection [44]. 

A total of 2 × 10^5^ cells per well were seeded in 24-well plates with PK15-WT, PK15-Rab4b-K.O., and PK15-Rab4b-O.E. cell lines to assess the effect of the *Rab4b* gene on JEV replication. Then, 12 h post-infection (hpi) and 24 h post-infection (hpi) RT–qPCR was performed in cells treated with the JEV (MOI = 0.5) when they were approximately 95% confluent.

Total RNA was extracted using an AxyPrep Multisource Total RNA extract Miniprep Kit (Axygen, San Francisco, CA, USA) from cells collected from 24-well cell culture plates. In reverse transcription, TransScript^®^ Uni All-in-One First-Strand cDNA Synthesis SuperMix (One-Step gDNA Removal) was used (Transgene, Beijing, China). Transgene’s Fast Green qPCR Super-Mix (Tli RNaseH Plus) (Transgene, Beijing, China) was used for RT–qPCR analysis of synthetic cDNA in a Bio-rad CFX96 System (Bio-rad, Hercules, CA, USA). Table 1 shows the primer sequences.

### 2.4. Assay for Indirect Immunofluorescence (IFA)

PBS was used to wash the cells three times, 4% paraformaldehyde was applied for one hour, and 0.1% Triton-X-100 was applied for 30 min to permeabilize the samples. Three washes with PBS, 4% paraformaldehyde, and 0.1% Triton-X-100 were performed to fix and permeabilize the cell samples, respectively. A primary anti-JEV envelope polyclonal antibody (Novus, Littleton, CO, USA) was diluted at 1:200 at 4 °C and a secondary Alexa Fluor 555-labeled Donkey Anti-Mouse IgG(H+L) antibody (Beyotime, Shanghai, China) was diluted at 1:200 at room temperature for 1h. Cells were stained with DAPI (Solarbio, Beijing, China) for 10 min and observed under an Olympus BX63 fluorescence microscope (Olympus, Tokyo, Japan).

### 2.5. Virus Adsorption Assay

For the analysis of the initial steps of virus replication, a virus adsorption assay was performed as follows. PK15-WT, PK15-Rab4b-K.O., and PK15-Rab4b-O.E. cells were seeded at 2 × 10^5^ cells per well in 24-well plates. When the cells were approximately 95% confluent, they were infected with the JEV (MOI = 10) at 4 °C for 1.5 h, at which temperature the virus can adsorb onto the cell surface but not internalize [45]. The cells were washed three times with PBS prepared for IFA and RT–qPCR assay.

### 2.6. Viral Internalization Assay

Viral internalization was detected as described previously [46,47,48]. PK15-WT, PK15-Rab4b-K.O., and PK15-Rab4b-O.E. cells were seeded in 24-well plates (2 × 10^5^ cells/well). When the cells were approximately 95% confluent, the cells were pretreated with the JEV (MOI = 10) in 500 μL of DMEM at 4 °C for 1.5 h. The unbound JEV particles were subsequently removed through three PBS washes, after which the cells were moved to 37 °C. After 1.5 h of incubation, the cells were washed with PBS and treated with proteinase K (1 mg/mL) (Invitrogen, Carlsbad, CA, USA) for 10 min at room temperature to remove the adsorbed but not internalized virus. The proteinase K was then inactivated with 2 mM phenylmethylsulfonyl fluoride (PMSF; Solarbio, Beijing, China) in PBS with 3% bovine serum albumin (BSA; Solarbio, Beijing, China). The cells were then washed three times with PBS prepared for IFA and RT–qPCR assay.

### 2.7. The Effect of Rab4b on JEV Assembly and Release

Virus replication includes adsorption, internalization, uncoating, biosynthesis, assembly, and release. To further determine whether Rab4b affects the assembly and release of the JEV bypassing the steps of virus adsorption and internalization, JEV RNA transcription and RNA electroporation transfection were performed as previously described [39] and briefly described below.

By digesting pACYC-JEV-SA14/U14163 with XhoI and purifying them by phenol/chloroform extraction, linearized cDNAs were then transcribed in vitro using the T7 mMESSAGE MACHINE Kit (Ambion, Austin, TX, USA). The procedures were followed according to the instructions provided by the manufacturer. An aliquot of the RNA was stored at −80 °C after it was resolved in RNase-free water.

A Bio-Rad GenePulser Xcell (Bio-Rad, Hercules, CA, USA) was used to electroporate JEV RNA into PK15-WT and PK15-Rab4b-K.O. cells at 150 V for 10 ms, square wave, three pulses at 3 s intervals. The RNA was introduced via electroporation using 10 µg of RNA per 5 × 10^5^ cells. After a 10 min recovery at room temperature, the transfected cells were mixed with 2 mL pre-warmed DMEM containing 10% FBS. The cell seeding density was 2 × 10^5^ per well in a 24-well plate, and incubation was conducted with 5% CO_2_ at 37 °C. After three to four passages, the cell samples were analyzed using IFA and RT–qPCR assay. 

### 2.8. Western Blot

An ice-cold RIPA buffer containing 1 mM PMSF was used to lyse cell samples for 30 min. All samples were centrifuged and then measured for protein concentration with a BCA protein assay kit (Beyotime, Shanghai, China). SDS–polyacrylamide gel electrophoresis (SDS–PAGE) was carried out to separate the samples. Following electrotransfer, the gels were mounted on polyvinylidene difluoride (PVDF) membranes (Bio-Rad, Hercules, CA, USA). Then, 5% skimmed milk was blocked at room temperature for 2 h, and anti-JEV envelope and anti-GAPDH polyclonal antibodies (Proteintech, Wuhan, China) were incubated overnight at 4 °C. Two hours after incubation with goat anti-rabbit or mouse IgG-HRP secondary antibodies at room temperature, the membranes were cleaned with four washes of TBST (tris-buffered saline containing 0.05% Tween 20). As a final step, images were captured using ChemiDoc TM MP Imaging System (BioRad, Hercules, CA, USA), and Gel-Pro Analyzer Version 6.3 (Media Cybernetics, Rockville, MD, USA) was used to quantify the expression level of the total E protein.

### 2.9. Statistical Analysis 

Each experiment was performed in triplicate, and the results are expressed as mean ± standard deviation. Statistical analysis was conducted on the data using SPSS Statistics 28 (SPSS, Chicago, IL, USA) software and GraphPad Prism 10 software (GraphPad Software, Inc., La Jolla, CA, USA). Statistical comparisons between the two groups were performed using unpaired *t*-tests. *p* < 0.05 was considered to indicate a statistically significant difference. *p* < 0.05, 0.01, and 0.001 were indicated by asterisks *, **, and ***, respectively. ns means *p* > 0.05.

## 3. Results

### 3.1. Effect of Rab4b Gene on Cell Viability

To evaluate the impact of *Rab4b* gene knockout or overexpression on cell viability in PK-15 cell lines, we inoculated PK15-WT, PK15-Rab4b-K.O., and PK15-Rab4b-O.E. cells with the JEV (MOI = 0.1) in 96-well plates for 12 and 24 h. Cell viability was subsequently assessed using the CCK-8 assay. Concurrently, cell morphology was examined using an optical microscope. The results indicate that the *Rab4b* gene did not significantly affect the viability of the three PK15 cell lines (Figure 1A), nor did it result in notable differences in cell morphology (Figure 1B). Based on these findings, the PK15-WT, PK15-Rab4b-K.O., and PK15-Rab4b-O.E. cell lines are suitable candidates for further investigation into viral infection.

### 3.2. Effect of Rab4b on JEV Replication 

In PK15-WT, PK15-Rab4b-K.O., and PK15-Rab4b-O.E. cell lines, JEV infection with an MOI of 0.5 was studied at 12 and 24 hpi, and supernatants were collected to determine Rab4b’s effect on JEV replication. Based on Ct values analysis of the JEV *E* gene, it was discovered that the PK15-Rab4b-K.O. cell line was suppressed compared to the PK15-WT cell line in terms of JEV replication (Figure 2A). 

In Figure 2A, RT–qPCR analysis was performed on PK15-WT, PK15-Rab4b-K.O., and PK15-Rab4b-O.E. cells infected with the JEV for 12 h and 24 h, respectively. The JEV E gene Ct values in PK15-Rab4b-K.O. cells were significantly higher than those in the PK15-WT and PK15-Rab4b-O.E. cells. Furthermore, no significant difference was found between the PK15-WT and the PK15-Rab4b-O.E. cells (Figure 2A). *Rab4b* gene knockouts resulted in a significant decrease in JEV replication. Infected with the JEV for 24 h, PK15-WT, PK15-Rab4b-K.O., and PK15-Rab4b-O.E. cells all showed similar Western blotting results (Figure 2B), and the JEV E protein expression in PK15-Rab4b-K.O. cells was significantly lower than that in PK15-WT and PK15-Rab4b-O.E. cells. Briefly, JEV replication is inhibited in PK15 cells with Rab4b knockout.

### 3.3. Effect of Rab4b on JEV Adsorption

The viral *E* gene Ct values in PK15-WT, PK15-Rab4b-K.O., and PK15-Rab4b-O.E. cells were determined using RT–qPCR. The JEV *E* gene Ct value of PK15-Rab4b-K.O. cells was very significantly higher than that of the wild type (*p* < 0.01), and the JEV *E* gene Ct values of PK15-Rab4b-O.E. cells were significantly lower than those of the wild type (*p* < 0.05) (Figure 3A). Similarly, IFA results indicate that very weak and less intense red fluorescence could be observed following virus attachment in PK15-Rab4b-K.O. cells inoculated with the JEV than in PK15-WT cells and that thick and strong intense red fluorescence could be observed following virus attachment in PK15-Rab4b-O.E. cells (Figure 3B). This indicates that Rab4b, as a host factor, can facilitate the adsorption of the JEV into cells.

### 3.4. Effect of Rab4b on JEV Internalization

The effect of Rab4b on JEV internalization was based on the JEV adsorption assay. The viral *E* gene Ct values in PK15-WT, PK15-Rab4b-K.O., and PK15-Rab4b-O.E. cells were determined using RT–qPCR. There was no significant difference between the JEV *E* gene Ct values of PK15-WT and PK15-Rab4b-k.O. cells (*p* > 0.05), but the JEV *E* gene Ct value of PK15-Rab4b-O.E. cells was very significantly higher than that of the wild type (*p* < 0.01) (Figure 4A). Meanwhile, a similar effect of Rab4b over-expression promoting the JEV internalization was observed by the IFA assay. The IFA results show that there red fluorescence could be seldom be observed following virus internalization in PK15-Rab4b-K.O. cells inoculated with the JEV than in PK15-WT cells, and thick and strong intense red fluorescence could be observed following virus internalization in PK15-Rab4b-O.E. cells (Figure 4B). It was also demonstrated that Rab4b could promote the internalization of the JEV into cells.

### 3.5. Effect of Rab4b on JEV Assembly and Release

To further determine whether Rab4b affects the assembly and release of the JEV, we conducted in vitro transcription and electroporation, skipping the step of virus adsorption and internalization, and directly introduced JEV RNA for corresponding detection.

First, the full-length infectious clone plasmid pACYC-JEV-SA14/U14163 was linearized using the *XhoI* restriction enzyme, and the template was purified by the phenol–chloroform method. The schematic of the JEV full-length cDNA clone of SA14/U14163 is shown in Figure 5A. Then, the T7 mMESSAGE MACHINE Kit was used for in vitro transcription. After transcription, phenol–chloroform was used for purification. Finally, 10 μg of in vitro synthesized JEV RNA template was introduced into PK15 cells by electroporation and cultured at 37 °C with 5% CO_2_. After electroporation, the cells can grow normally, and there is no contamination, indicating that the electroporation process was successful and can provide biological materials for further identification of RNA introduction and JEV packaging (Figure 5B,C). As Figure 5B shows, the plasmids of the two groups were fully digested without extra bands, and after restriction enzyme *XhoI* digestion, the plasmid size was above the 15,000 bp indicator band of the DNA marker, which was consistent with the expected size. Figure 5C shows that PK15-WT and PK15-Rab4b-K.O. cells grew normally and were not contaminated. All the cell morphology showed a normal spindle shape before and after electroporation. This demonstrated that the electroporation was successful. However, PK15-WT and PK15-Rab4b-K.O. cells grew slowly, and the degree of cell fusion was not high at 24 h after electroporation, so the cells’ culture time needed to be appropriately extended (about 3 days) for the next experiment. 

Next, we used PK15-WT and PK15-Rab4b-K.O. cells that were seeded at 2 × 10^5^ cells per well in 24-well plates after 3 days of culture and detected the success of JEV packaging by IFA, RT–qPCR, and Western blot. The results show that PK15-WT and PK15-Rab4b-K.O. cells could successfully produce JEV virus particles after electroporation (Figure 6A–D). According to the IFA results, PK15-WT and PK15-Rab4b-K.O. cells can both form virus particles, but the virus particles in the wild type are more than those in the PK15-Rab4b-K.O. cells (Figure 6A). The JEV *E* gene Ct values of the PK15-Rab4b-K.O. cells were very significantly higher than those of the wild type (*p* < 0.01) (Figure 6B). The Western blot results (Figure 6C) and the relative total JEV E protein expression level of PK15-Rab4b-K.O. cells were extremely significantly lower than for the wild type (*p* < 0.001) (Figure 6D). These results indicate that the lack of Rab4b inhibits the assembly and release of the JEV.

## 4. Discussion

Virus replication depends on host cell function, as we know. Hence, it is imperative to understand the role of virus–host interactions during virus replication in order to develop novel anti-virus drugs. Numerous studies have revealed various host factors associated with the JEV that operate through distinct mechanisms [18,49,50,51,52]. CRISPR-Cas9 screening is a powerful, high-throughput tool for identifying common host factors critical for virus propagation [53]. Through the use of a genome-wide CRISPR/Cas9 knockout library in PK15 cells, we identified Rab4b as the host factor associated with JEV replication. In this study, we used PK15-WT, PK15-Rab4b-K.O., and PK15-Rab4b-O.E. cells to perform a series of studies around the life cycle of the JEV, and for the first time, our studies showed that Rab4b plays a critical role in JEV infection and replication in PK15 cells.

Small GTPases, Rabs (Ras-related in the brain), are critical in regulating intracellular vesicle transport. Research has indicated that Rab proteins play an important role in the replication and entry of the Hepatitis C Virus (HCV) [54], and that Rab5 is essential for the cellular entry of both dengue and West Nile viruses [55]. In 1995, Rab4b, a small GTPase found in the Rab family, was discovered. Since then, much has been learned about Rab4b’s structure and function [56]. It has been found that Rab4b proteins are co-regulated with MHC Class II genes [57], and Rab4b also interacts with early endosomes in the early stages of endocytosis [57,58]. Furthermore, Rab4b participates in cellular trafficking as well as inflammation and insulin response, particularly via T cells. As obesity affects T cells in adipose tissue, Rab4b levels may decrease, leading to insulin resistance and dysfunctional adipose tissue [59]. Rab4b is poorly understood in terms of its effects on the virus life cycle, with an emphasis on primary replication sites where the virus invades hosts. Only Turner et al. found that the disruption of Rab4b significantly limited human cytomegalovirus (HCMV) replication [60], and McCormick et al. found that, as HCMV infected Rab4b, it moved to the viral assembly compartment and reduced virion particle release, suggesting that Rab4b was involved in virion assembly and egress. This study not only further confirmed the impact of Rab4b on the proliferation of JEV, but also conducted an initial exploration into the role of Rab4b within the JEV life cycle from its perspective.

Host factors are broadly categorized as either dependency factors that promote viral replication or as restriction factors that inhibit viral replication and must be somehow counteracted for successful virus replication [61]. The viral life cycle of the JEV follows the usual basic stages of the enveloped virus, which can be divided into adsorption, internalization, viral genome replication, translation, assembly, and release stages. If host factors can influence any of these stages, they can significantly inhibit or enhance viral replication and control the spread of the virus. Therefore, in our study, we examined the impact of Rab4b on JEV proliferation across the stages of adsorption, internalization, assembly, and release within the JEV replication lifecycle.

We first characterize the effects of Rab4b on JEV replication and found that the knockout of *Rab4b* inhibits the replication of the JEV in PK15 cells. On the contrary, the overexpression of *Rab4b* promoted the replication of the JEV in PK15 cell lines.

In adsorption, the virus surface envelope protein (E) is responsible for JEV entry into host cells. During virus entry, the E protein of the JEV first binds to a receptor on the host cell surface for viral attachment and subsequently fuses viral and host membranes. However, the receptor for the JEV is still being investigated [62,63]. Therefore, we particularly focused on the initial phase of virus adsorption. RT–qPCR and IFA methods were used, and JEV adsorption was also analyzed. The virus adsorption assay results show that Rab4b, as a host factor, can facilitate the adsorption of the JEV into cells at the gene or protein levels. From the data in the UniPort database (www.uniprot.org, accessed on 24 July 2024), we discovered that the Rab4b protein is localized on the cell membrane. Does this imply that the JEV particles directly bind to Rab4b to enter the cell? Furthermore, research has shown that Rab5, a protein closely related to Rab4, strongly co-localizes with viral particles during flavivirus invasion [64,65]. Thus, it is hypothesized that the Rab4b protein may co-localize with, or interact directly with, viral particles, thereby affecting their adsorption.

In the internalization stage, our results demonstrate that Rab4b could promote the internalization of the JEV into cells by RT–qPCR and IFA methods. This result is similar to that of previous related studies, such as the GTPases Rab5 and Rab7 control the highly dynamic endocytic network that the budded virus of *Autographa californica* multiple nucleopolyhedrovirus (AcMNPV) particles enter after internalization into the cytoplasm [66]. Rab5 has been implicated in the internalization of HIV-1 [67].

In the assembly and release stage, to further determine whether Rab4b affects the assembly and release of JEV, we conducted in vitro transcription and electroporation, skipping the step of virus adsorption and internalization, and directly introduced JEV RNA for corresponding detection. Our results indicate that the lack of Rab4b inhibited the assembly and release of the JEV. Although there are no relevant reports on the assembly and release of Rab4b with other viruses, our result is similar to that of previous related studies which have shown that the knockdown of three Rab GTPases, Rab8a, Rab10, and Rab11a, promoted viral accumulation and suppressed viral release, indicating that the Epstein–Barr Virus (EBV) utilizes the host cellular secretory pathway to support progeny virion release [68].

Understanding the mechanisms underlying virus–host interactions is of paramount importance for elucidating viral infection and pathogenesis, as well as for the development of antiviral therapeutics. This constitutes a fundamental scientific issue within the field of virology. Host proteins are increasingly being recognized as potential targets for antiviral drug development across various stages of the viral life cycle [69,70]. Our initial findings indicate that the host factor Rab4b facilitates the replication of the Japanese encephalitis virus (JEV) throughout its life cycle, encompassing the stages of adsorption, internalization, assembly, and release. Therefore, our findings provide a critical foundation for the future development of Rab4b as a therapeutic target for anti-JEV drugs.

In conclusion, our study demonstrated for the first time that host factor Rab4b facilitates the adsorption, internalization, assembly, and release of the JEV, thereby promoting JEV replication. However, the precise mechanism by which Rab4b functions still requires further investigation. 

## 5. Conclusions

Our previous laboratory work identified Rab4b as a host factor involved in JEV replication using CRISPR library screening. In this study, we used PK15-WT, PK15-Rab4b-K.O., and PK15-Rab4b-O.E. cells, along with CCK-8, IFA, RT–qPCR, and Western blot assays, to investigate Rab4b’s role in the JEV’s life cycle. We discovered that Rab4b promotes JEV replication, influencing stages such as adsorption, internalization, assembly, and release. This study establishes a foundational basis for the in-depth investigation of the specific mechanisms through which Rab4b modulates the progression of JEV infection. Furthermore, understanding the role of Rab4b in JEV replication will enhance the elucidation of the interaction network between the JEV and host proteins, deepen our comprehension of the JEV replication cycle, and identify potential targets for the development of novel anti-JEV therapeutics.

## Figures and Tables

**Figure 1 microorganisms-12-01804-f001:**
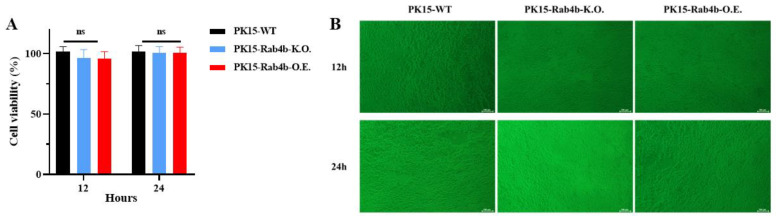
Assessing cell viability and morphology using CCK-8 assay and light microscopy. (**A**) To assess the cell growth, PK15-WT, PK15-Rab4b-K.O., and PK15-Rab4b-O.E. cells were plated onto 96-well plates and cell viability was determined using the CCK-8 assay (*n* = 6 experimental replicates, ns means *p* > 0.005). (**B**) Typical morphology of PK15-WT, PK15-Rab4b-K.O., and PK15-Rab4b-O.E. cells cultured in DMEM medium for 12 h and 24 h (100×, scale bar = 100 µm).

**Figure 2 microorganisms-12-01804-f002:**
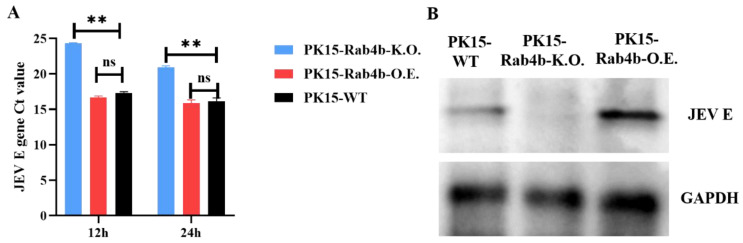
The impact of Rab4b on JEV replication. (**A**) PK15-WT, PK15-Rab4b-K.O., and PK15-Rab4b-O.E. cells were infected by JEV after 12 and 24 hpi, and quantified by RT–qPCR to obtain the JEV *E* gene Ct values. (**B**) The Western blot results for the PK15-WT, PK15-Rab4b-K.O., and PK15-Rab4b-O.E. cells after infection by JEV 24 hpi. (** means *p* < 0.01, ns means *p* > 0.05).

**Figure 3 microorganisms-12-01804-f003:**
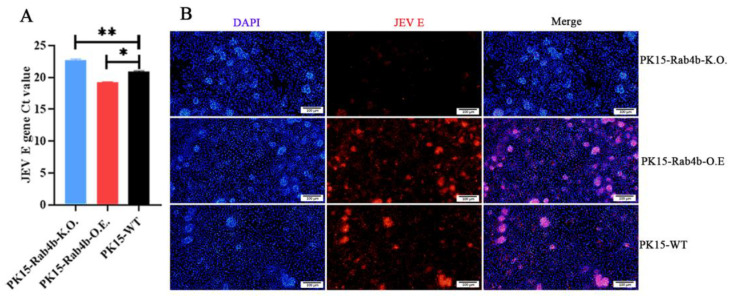
The effect of Rab4b on JEV adsorption. (**A**) PK15-WT, PK15-Rab4b-K.O., and PK15-Rab4b-O.E. cells were infected with JEV (MOI = 10) at 4 °C for 1.5 h; then, the cell samples were washed three times with PBS, and quantified by RT–qPCR to obtain the JEV *E* gene Ct values (* means *p* < 0.05, ** means *p* < 0.01). (**B**) The IFA results for the PK15-WT, PK15-Rab4b-K.O., and PK15-Rab4b-O.E. cells infected with JEV (MOI = 10) at 4 °C for 1.5 hpi. Red fluorescence indicates the JEV E-protein. (200×, scale bar = 100 µm).

**Figure 4 microorganisms-12-01804-f004:**
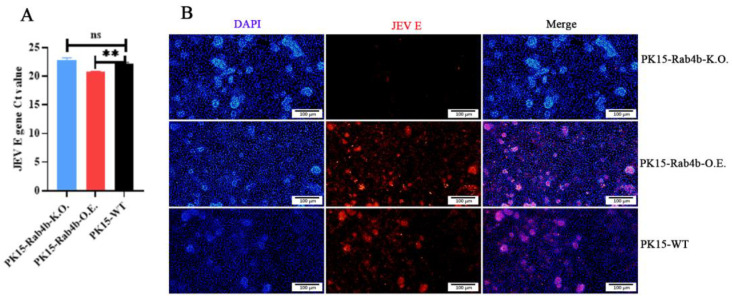
The effect of Rab4b on JEV internalization. PK15-WT, PK15-Rab4b-K.O., and PK15-Rab4b-O.E. cells were infected with JEV (MOI = 10) in 500 μL of DMEM at 4 °C for 1.5 h, and the unbound JEV particles were subsequently removed through three PBS washes, after which the cells were moved to 37 °C. After 1.5 h of incubation, the cells were washed with PBS and treated with proteinase K (1 mg/mL) for 10 min at room temperature to remove the adsorbed but not internalized virus. The proteinase K was then inactivated with 2mM phenylmethylsulfonyl fluoride (PMSF) in PBS with 3% bovine serum albumin (BSA). The cells were then washed three times with PBS prepared for IFA and RT–qPCR assay. (**A**) PK15-WT, PK15-Rab4b-K.O., and PK15-Rab4b-O.E. cells quantified by RT–qPCR to obtain the JEV *E* gene Ct values (** means *p* < 0.01, ns means *p* > 0.05). (**B**) The IFA results for the PK15-WT, PK15-Rab4b-K.O., and PK15-Rab4b-O.E. cells. Red fluorescence indicates the JEV E-protein. (200×, scale bar = 100 µm).

**Figure 5 microorganisms-12-01804-f005:**
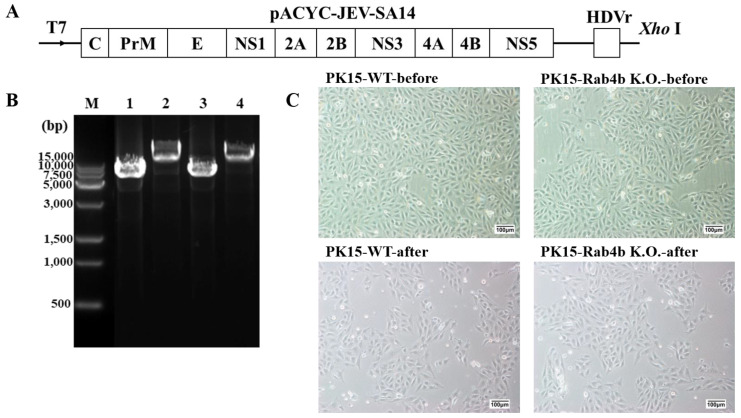
Linearization identification of a full-length infected clone plasmid containing JEV SA14 genome and in vitro transcription and electroporation of JEV RNAs. (**A**) The schematic of the JEV full-length cDNA clone of SA14/U14163. (**B**) Linearization identification of a full-length infected clone plasmid containing JEV SA14 genome. Lane M, DNA marker; Lane 1 and Lane 3, control plasmid; Lane 2 and Lane 4, the fragments obtained by linearization of plasmid with restriction enzyme *XhoI*. (**C**) The upper row in the figure shows the cell growth of PK15-WT and PK15-Rab4b-K.O. cultured in the cell culture flasks for 24 h before electroporation, and the lower row in the figure shows the cell growth of PK15-WT and PK15-Rab4b-K.O. cultured in the 6-well plate for 24 h after electroporation (200×, scale bar = 100 µm).

**Figure 6 microorganisms-12-01804-f006:**
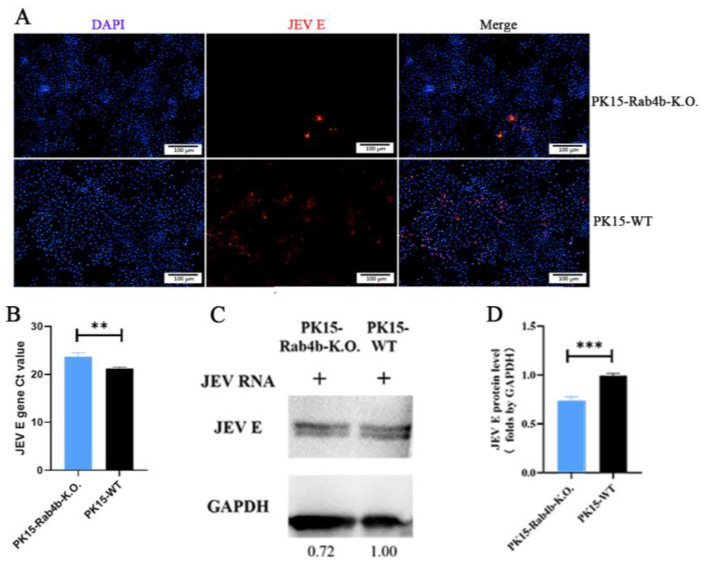
The effect of Rab4b on JEV assembly and release. (**A**) IFA of PK15-WT and PK15-Rab4b-K.O. cells after 3 days in vitro transcription and electroporation of JEV RNAs. Red fluorescence indicates the JEV E-protein. (200×, scale bar = 100 µm). (**B**) RT–qPCR quantified the JEV E gene Ct values of PK15-WT and PK15-Rab4b-K.O. cells. (**C**) The Western blot results for the PK15-WT and PK15-Rab4b-K.O. cells after 3 days of in vitro transcription and electroporation of JEV RNAs. (**D**) The relative total JEV E protein expression level was determined by the intensity of Western blot bands with the software Gel-Pro Analyzer Version 6.3 (Media Cybernetics). (** means *p* < 0.01, *** means *p* < 0.001).

**Table 1 microorganisms-12-01804-t001:** Primer sequences for RT–qPCR.

Target Names	(5′→3′) Primer Sequences
JEV-E1223-F	CAGTGGAGCCACTTGGGTG
JEV-E1223-R	TTGTGAGCTTCTCCTGTCG
GAPDH-PF	TATGATTCCACCCACGGCAAG
GAPDH-PR	ATACGTAGCACCAGCATCACC

## Data Availability

The novel findings and contributions of this research are detailed in the published manuscript. Further questions regarding the study can be addressed to the authors’ listed correspondence.

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
