# Peer review of "Host Factor Rab4b Promotes Japanese Encephalitis Virus Replication"

_microorganisms, 2024, doi:10.3390/microorganisms12091804_

Round 1

Reviewer 1 Report

Comments and Suggestions for Authors

Comments and suggestions

Summary section:

1. The first sentence is redundant

Introduction section:

2. Indicate examples of currently known target cells and receptors.

Introduction section:

3. Indicate the vectors that transmit the virus

4. Clearly indicate the objective of the study at the end of the introduction

Materials and methods section:

5. Develop an algorithm of the methodology for greater clarity.

Results section:

6. Restructure the first paragraph for better understanding

Conclusions section:

7. The conclusions must be based on the objectives and the results.

Comments on the Quality of English Language

 Minor editing of English language required.

Author Response

Author's Reply to the Review Report Reviewer 1

Comments 1: Summary section: The first sentence is redundant

Response 1: Thank you for your advice.  We agree and have cut the first sentence to make the text more succinct and less repetitive. We have also carefully revised the abstract section to improve the quality of this manuscript.

Please refer to lines 21-32 of the revised manuscript for the updated Abstract section.

Comments 2: Introduction section: Indicate examples of currently known target cells and receptors.

Response 2: Thank you for your advice. We have incorporated the suggested addition as follows.

For the JEV invasive body, different cell types, attachment factors, and receptors have been identified. Several molecules have been proposed as cell receptors to JEV, including Heparan sulfate glycosaminoglycans (HSPG), HSP70, and HSP90β, vimentin, CD4, laminin receptor, and α5β3 integrin [1]. Further research disclosed that Glycosaminoglycans (GAGs) / glucose-regulated protein 78 (GRP78) / HSPG act as attachment factors for JEV in BHK-21 cells, C6/36 mosquito cells, CHO-K1 (derived from Chinese hamster ovary) cells, mouse neuronal (Neuro2a) cells, mouse primary neurons, and human epithelial Huh-7 cells [2,3].  HSP90β was demonstrated as a receptor for JEV in Vero cells [4]. PK15 cell's and BHK-21 cell’s surface vimentin has proven to be the receptor for JEV entry and interacts with the E protein of JEV [5]. Low-density lipoprotein receptor (LDLR) is a possible cellular receptor for JEV since it is involved in JEV entry into the A549 cells and can bind to JEV-E [6]. The fractalkine receptor CX3CR1 involved the JEV infection by exposure to live JEV and the vaccine in human microglia[7]. RIPK3 promotes JEV replication in neurons by downregulating IFI44L [8]. Although previous studies have identified the target cells and receptors for JEV infection, the detailed molecular mechanism for JEV entry into host cells remains unclear[9,10].
        Please refer to lines 128-142 of the revised manuscript for the updated Introduction section.

Comments 3: Introduction section: Indicate the vectors that transmit the virus.

Response 3: Thank you for your advice. We have incorporated the suggested addition as follows.

JEV is mainly transmitted by Culex tritaeniorhynchus in regions where the virus is endemic, but other mosquito species, especially within the genus Culex, can act as vectors [11]. In addition, Field surveys and laboratory assessments indicate that Ar. subalbatus is a major vector of JEV transmission [12]. Typically, human infection is caused by Culex species infected by JEV, transmitted in enzootic cycles between mosquitoes, pigs, and birds [13].

        Please refer to lines 117-122of the revised manuscript for the updated Introduction section.

Comments 4: Introduction section: Clearly indicate the objective of the study at the end of the introduction.

Response 4: Thank you for your advice. This has been modified as recommended.

Please refer to lines 159-167 of the revised manuscript for the updated Introduction section.

Comments 5: Materials and methods section: Develop an algorithm of the methodology for greater clarity.

Response 5: Thank you for your advice. We have modified ”Materials and methods section” as recommended.

Please refer to lines 383-726 of the revised manuscript for the updated Materials and methods section.

Comments 6: Results section: Restructure the first paragraph for better understanding.

Response 6: Thank you for your advice. We have restructured the first paragraph as your suggested.

Please refer to lines 729-737 of the revised manuscript for the updated Results section.

Comments 7: Conclusions section: The conclusions must be based on the objectives and the results.

Response 7: Thank you for your advice. We have checked and revised the Conclusion section as your suggested.

Please refer to lines1257-1267 of the revised manuscript for the updated Conclusion section.

Comments 8: Comments on the Quality of English Language Minor editing of English language required.

Response 8: Thank you for the positive comments and highlighting areas for improvement. In the whole new manuscript, we have improved the English with the help of a native speaker and professional editor.

References for this response

  1. María-Angélica Calderón-Peláez , M.L.V.-R., Leidy Y Bastidas-Legarda , Edgar O Beltrán, Sigrid J Camacho-Ortega, Jaime E Castellanos. Dengue Virus Infection of Blood-Brain Barrier Cells: Consequences of Severe Disease Frontiers In Microbiology 2019, 10, 1435, doi:doi: 10.3389/fmicb.2019.01435. .
  2. Yu-Jung Chien, W.-J.C., Wei-Li Hsu, Shyan-Song Chiou. Bovine lactoferrin inhibits Japanese encephalitis virus by binding to heparan sulfate and receptor for low density lipoprotein Virology 2008, 379, 143-151, doi:10.1016/j.virol.2008.06.017.
  3. Minu Nain, S.M., Sonali Porey Karmakar, Adrienne W Paton, James C Paton, M Z Abdin, Anirban Basu, Manjula Kalia, Sudhanshu Vrati. GRP78 Is an Important Host Factor for Japanese Encephalitis Virus Entry and Replication in Mammalian Cells. Journal of Virology 2017, 91, e02274-02216, doi:10.1128/JVI.02274-16.
  4. Yuan Wang, Y.L., Tianbing Ding. Heat shock protein 90β in the Vero cell membrane binds Japanese encephalitis virus International Journal of Molecular Medicine 2017, 40, 474-482, doi:10.3892/ijmm.2017.3041.
  5. Peng Wang, X.L., Qi Li, Jue Wang, Wenke Ruan Proteomic analyses identify intracellular targets for Japanese encephalitis virus nonstructural protein 1 (NS1) Virus Research 2021, 302, 198495, doi:doi: 10.1016/j.virusres.2021.198495. .
  6. Lihong Huang, H.L., Zuodong Ye,Qiang Xu, Qiang Fu,Wei Sun, Wenbao Qi, Jianbo Yue. Berbamine inhibits Japanese encephalitis virus (JEV) infection by compromising TPRMLs-mediated endolysosomal trafficking of low-density lipoprotein receptor (LDLR). Emerging Microbes and Infections 2021, 10, 1257-1271, doi:10.1080/22221751.2021.1941276.
  7. Nils Lannes, V.N., Brigitte Scolari, Solange Kharoubi-Hess, Michael Walch, Artur Summerfield, Luis Filgueira. Interactions of human microglia cells with Japanese encephalitis virus. Virology Journal 2017, 14, 8, doi:10.1186/s12985-016-0675-3. .
  8. Bian, P.; Ye, C.; Zheng, X.; Luo, C.; Yang, J.; Li, M.; Wang, Y.; Yang, J.; Zhou, Y.; Zhang, F.; et al. RIPK3 Promotes JEV Replication in Neurons via Downregulation of IFI44L. Frontiers in Microbiology 2020, 11, doi:10.3389/fmicb.2020.00368.
  9. Anwar, M.N.; Akhtar, R.; Abid, M.; Khan, S.A.; Rehman, Z.U.; Tayyub, M.; Malik, M.I.; Shahzad, M.K.; Mubeen, H.; Qadir, M.S.; et al. The interactions of flaviviruses with cellular receptors: Implications for virus entry. Virology 2022, 568, 77-85, doi:10.1016/j.virol.2022.02.001.
  10. Chiou, S.S.; Liu, H.; Chuang, C.K.; Lin, C.C.; Chen, W.J. Fitness of Japanese encephalitis virus to Neuro-2a cells is determined by interactions of the viral envelope protein with highly sulfated glycosaminoglycans on the cell surface. J Med Virol 2005, 76, 583-592, doi:10.1002/jmv.20406.
  11. Luis M. Hernández-Triana, A.J.F., Sanam Sewgobind, Fabian Z. X. Lean, Stuart Ackroyd, Alejandro Nuñez, Sarah Delacour, Andrea Drago, Patrizia Visentin, Karen L. Mansfield,Nicholas Johnson. Susceptibility of Aedes albopictus and Culex quinquefasciatus to Japanese encephalitis virus. Parasites & Vectors 2022, 15, 210, doi:10.1186/s13071-022-05329-0.
  12. Chun-xiao Li, X.-x.G., Yong-qiang Deng, Qin-mei Liu, Dan Xing, Ai-juan Sun, Qun Wu, Yan-de Dong, Ying-mei Zhang, Heng-duan Zhang, Wu-chun Cao, Cheng-feng Qin, Tong-yan Zhao. Susceptibility of Armigeres subalbatus Coquillett (Diptera: Culicidae) to Zika virus through oral and urine infection. PLoS Neglected Tropical Diseases 2020, 14, e0008450, doi:10.1371/journal.pntd.0008450.
  13. Tehmina Bharucha, B.C., Alice Farmiloe, Elizabeth Sutton,  Hanifah Hayati, Peggy Kirkwood,  Layal Al Hamed, Nadja van Ginneken,  Krishanthi S. Subramaniam, Nicole Zitzmann,  Gerry Davies,  Lance Turtle. Mouse models of Japanese encephalitis virus infection: A systematic review and meta-analysis using a meta-regression approach. PLoS Neglected Tropical  Diseases 2022, 16, e0010116, doi:10.1371/journal.pntd.0010116.

Reviewer 2 Report

Comments and Suggestions for Authors

Minor comments

The introduction provides an adequate general background on JEV and the role of Rab proteins in viral replication. However, it does not sufficiently justify the choice of the PK15 cell model nor the importance of explicitly studying Rab4b in the context of JEV.

The discussion touches on key points of the study but does not sufficiently explore the implications of the findings for the development of targeted antiviral therapies.

Major comments

It would be interesting if the authors analyzed the subcellular localization of Rab4b during JEV infection. Multiple host proteins have been described as relocalizing during flavivirus infection to participate in viral infection.

It would be interesting if the authors performed IFA co-localization assays to analyze the subcellular localization of Rab4b in comparison to specific viral proteins (such as C, M, or E) in infected cells. The co-localization of Rab4b with specific viral components could provide visual evidence of its role in viral replication and its potential as a therapeutic target.

IFI images should be improved by removing the extranuclear DAPI background (where it does not correspond), observing the images with greater detail and quality, and adding the size bars.

Author Response

Author's Reply to the Review Report Reviewer 2

Minor comments

Comments 1: The introduction provides an adequate general background on JEV and the role of Rab proteins in viral replication. However, it does not sufficiently justify the choice of the PK15 cell model nor the importance of explicitly studying Rab4b in the context of JEV.

Response 1:  Thank you for the positive comments and highlighting areas for improvement. PK15 is a cell line exhibiting epithelial morphology that was isolated from the kidney of an adult pig, and PK15 cells are a widely utilized model for the propagation of JEV [1-5]. We added this in the revised Introduction section. Given their suitability for JEV culture and infection, we used CRISPR/Cas9 whole-genome library screening to identify Rab4b as a critical host factor that enhances JEV replication in PK15 cells. Notably, PK15 cells are derived from porcine kidney tissue, which is consistent with the fact that pigs serve as the primary hosts for Japanese Encephalitis Virus (JEV). Furthermore, Rab4b is ubiquitously expressed across a range of mammalian organs, tissues, and cells, including PK15 cell lines. Consequently, this study employed PK15 cells as a model system to elucidate the mechanisms underlying JEV replication.

Please refer to lines 21-32 of the revised manuscript for the updated Introduction section.

Comments 2: The discussion touches on key points of the study but does not sufficiently explore the implications of the findings for the development of targeted antiviral therapies.

Response 2: Thank you for the positive comments and highlighting areas for improvement. Based on your recommendation, we have expanded the discussion section, with particular emphasis on a comprehensive analysis of the relationship between the research findings and antiviral treatment.

Understanding the mechanisms underlying virus-host interactions is of paramount importance for elucidating viral infection and pathogenesis, as well as for the development of antiviral therapeutics. This constitutes a fundamental scientific issue within the field of virology. Host proteins are increasingly being recognized as potential targets for antiviral drug development across various stages of the viral life cycle[6,7]. Our initial findings indicate that the host factor Rab4b facilitates the replication of Japanese Encephalitis Virus (JEV) throughout its life cycle, encompassing the stages of adsorption, internalization, assembly, and release. Therefore, our findings provide a critical foundation for the future development of Rab4b as a therapeutic target for anti-JEV drugs.

Please refer to lines 1243-1251 of the revised manuscript for the updated Discussion section.

Major comments

Comments 3: It would be interesting if the authors analyzed the subcellular localization of Rab4b during JEV infection. Multiple host proteins have been described as relocalizing during flavivirus infection to participate in viral infection.

Response 3: Thank you for the positive comments and highlighting areas for improvement.

Yes, analyzing the subcellular localization of Rab4b during JEV infection will help uncover its mechanism. Our lab is already conducting this research to explore the interaction sites between Rab4b and JEV more deeply. We appreciate your valuable suggestion.

Comments 4: It would be interesting if the authors performed IFA co-localization assays to analyze the subcellular localization of Rab4b in comparison to specific viral proteins (such as C, M, or E) in infected cells. The co-localization of Rab4b with specific viral components could provide visual evidence of its role in viral replication and its potential as a therapeutic target.

Response4: Thank you for the positive comments and highlighting areas for improvement.

Indeed, as per your suggestion, it would be highly informative if the authors conducted immunofluorescence assay (IFA) co-localization analysis to investigate the subcellular localization of Rab4b and specific viral proteins (such as C, M, or E) in infected cells. We fully acknowledge and appreciate your valuable input and intend to pursue comprehensive research in this area in the future.

Comments 5: IFI images should be improved by removing the extranuclear DAPI background (where it does not correspond), observing the images with greater detail and quality, and adding the size bars.

Response 5: Thank you for the positive comments and highlighting areas for improvement. As per your suggestion, the quality of all images has been improved. 

Please refer to Figure3B, Figure 4B and Figure 6A of the revised manuscript for the updated Results section.

References for this response

  1. Songbai Yang, M.H., Xiangdong Liu, Xinyun Li,1 Bin Fan, Shuhong Zhao. Japanese encephalitis virus infects porcine kidney epithelial PK15 cells via clathrin- and cholesterol-dependent endocytosis. Virology Journal 2013, 10, 258, doi:doi: 10.1186/1743-422X-10-258. .
  2. Jichen Niu, Y.J., Hao Xu, Changjing Zhao, Guodong Zhou, Puyan Chen, Ruibing Cao. TIM-1 Promotes Japanese Encephalitis Virus Entry and Infection. Viruses 2018, 10, 630, doi: 10.3390/v10110630.
  3. Kumari Chandan, M.G., Maryam Sarwat. Role of Host and Pathogen-Derived MicroRNAs in Immune Regulation During Infectious and Inflammatory Diseases. Frontiers In Immunology 2020, 10, 3081, doi:doi: 10.3389/fimmu.2019.03081. eCollection 2019.
  4. Weimin Xu, K.Y., Yi Zheng, Sanjie Cao, Qigui Yan,  Xiaobo Huang, Yiping Wen, Qin Zhao,Senyan Du, Yifei Lang, Shan Zhao, Rui Wu. BAK-Mediated Pyroptosis Promotes Japanese Encephalitis Virus Proliferation in Porcine Kidney 15 Cells. Viruses 2023, 15, 974, doi:doi: 10.3390/v15040974.
  5. Xiaolong Zhou, Q.Y., Chen Zhang, Zhenglie Dai, Chengtao Du, Han Wang, Xiangchen Li, Songbai Yang, Ayong Zhao. Inhibition of Japanese encephalitis virus proliferation by long non-coding RNA SUSAJ1 in PK-15 cells. Virology Journal 2021, 18, 29, doi:doi: 10.1186/s12985-021-01492-5.
  6. Sovan Saha, P.C., Anup Kumar Halder, Mita Nasipuri,Subhadip Basu, Dariusz Plewczynski. ML-DTD: Machine Learning-Based Drug Target Discovery for the Potential Treatment of COVID-19. Vaccines (Basel) 2022, 10, 1643, doi:doi: 10.3390/vaccines10101643.
  7. Catherine A. Freije, P.C.S. Detect and destroy: CRISPR-based technologies for the response against viruses. Cell Host & Microbe 2021, 29, 689-703, doi:doi: 10.1016/j.chom.2021.04.003.

Reviewer 3 Report

Comments and Suggestions for Authors

I think that the subject is interesting, it brings new elements in the way of JEV replication.

Compared to those studies, it also offers a perspective on JEV replication?

Knowing the role of Rab4b in JEV replication in pigs, what do you think are the prospects for the human population, considering that humans are not a natural host for JEV?

The conclusions are clear.

The references are relevant and cover the topic addressed.

I also have some questions for the authors:

What is the incidence of reported cases of JEV infection globally and in China?

If there is a real ratio or are the cases of JEV encephalitis possibly underreported?

If there is a difference in terms of incidence and mortality in adults and children?

What are the main clinical manifestations of JEV and what are their sequelae, to emphasize the importance of preventing this encephalitis!

What is the life cycle of JEV??

What are or could be the future treatment perspectives, knowing the role of Rab4b in JEV replication??

Author Response

Author's Reply to the Review Report Reviewer 3

Comments and Suggestions for Authors

Comments 1: I think that the subject is interesting, it brings new elements in the way of JEV replication.

Response 1:  Thank you for the positive comments and support.

Comments 2: Compared to those studies, it also offers a perspective on JEV replication?

Response 2:  Thank you for the positive comments and highlighting areas for improvement. Yes, we are confident that our research offers a novel perspective on the replication mechanisms of JEV.

Comments 3: Knowing the role of Rab4b in JEV replication in pigs, what do you think are the prospects for the human population, considering that humans are not a natural host for JEV?

Response 3: Thank you for the positive comments and highlighting areas for improvement. From an academic perspective, elucidating the role of Rab4b in porcine JEV replication may yield novel targets and strategies for the prevention and control of Japanese encephalitis in pigs. Given that pigs are pivotal in the epidemiology of Japanese encephalitis and humans are not the natural hosts of JEV, mitigating the incidence of the disease in pigs could alleviate the economic burden on the swine industry and concurrently reduce the transmission risk to humans. This has substantial implications for public health and safety.  On the other hand, the Rab4b protein has been detected in a variety of tissues, cells, and organelles across numerous mammalian species, including humans. The present study, which investigates the role of Rab4b in porcine models concerning JEV replication, holds significant implications for future research and the development of antiviral therapeutics targeting JEV in humans.

Comments 4: The conclusions are clear.

Response 4:  Thank you for the positive comments and support.

Comments 5: The references are relevant and cover the topic addressed.

Response 5:  Thank you for the positive comments and support.

Comments 6: What is the incidence of reported cases of JEV infection globally and in China?

Response 6:  Thank you for the positive comments and highlighting areas for improvement. According to the most recent reports from July 2024 [1], JEV identified as one of the primary etiological agents of viral encephalitis, with an estimated annual incidence ranging from 30,000 to 50,000 cases. JEV is considered endemic in at least 24 countries across Asia and Oceania, with approximately 3 billion individuals residing in regions affected by JEV epidemics. The virus is classified into five genotypes, with genotype I being the most prevalent. In recent years, the emergence of genotype IV cases in Australia and genotype V cases in South Korea has raised concerns regarding the pathogenicity of these strains and the extent of cross-protection provided by JEV vaccines against various genotypes [2]. In China, the incidence of JEV infections has markedly decreased as a result of widespread vaccination efforts. Specifically, the number of reported cases in 2021 and 2022 were 209 and 153, respectively [3].

Comments 7: If there is a real ratio or are the cases of JEV encephalitis possibly underreported?

Response 7: Thank you for the positive comments and highlighting areas for improvement. We conducted literature search and consulted some experts in public health. As we know, Japanese encephalitis, being a zoonotic disease, is subject to stringent prevention and control measures in China. Upon the identification of confirmed or suspected cases, reporting is conducted in strict adherence to regulatory frameworks, including the Law of the People's Republic of China on the Prevention and Control of Infectious Diseases, the Measures for the Management of Information Reporting on Public Health Emergencies and Infectious Disease Outbreak Monitoring, and the National Standards for the Management of Public Health Emergency Information Reporting (Trial). So there will be no concealment.

    Due to variations in infectious disease policies and management practices across different countries and regions, the reporting of Japanese encephalitis cases may be incomplete.

Comments 8: If there is a difference in terms of incidence and mortality in adults and children?

Response 8: Thank you for the positive comments and highlighting areas for improvement.

There is a difference between adult and children in terms of incidence, children and older adult seems more susceptibility. Japanese encephalitis is the most common cause of encephalitis in Asia, affecting mainly children [4].For example, in outbreak of indigenous genotype 4 JEV in Australia, a total of 21 cases had clear age information. That showed the affected people were aged between 20 to 70 years, except for three pediatric cases (12.38%, 3/21). At least eight cases (38.1%, 8/21) among individuals who were aged 60 and above[2].

Comments 9: What are the main clinical manifestations of JEV and what are their sequelae, to emphasize the importance of preventing this encephalitis!

Response 9: Thank you for the positive comments and highlighting areas for improvement. We have incorporated the suggested addition as follows.

In humans, the primary clinical manifestations and sequelae of Japanese Encephalitis predominantly encompass fever, convulsions, and alterations in sensory organ function. [5]. Among those who develop the disease, approximately one-third recover completely, one-third experience severe lifelong neurological complications, and one-third succumb to the illness. Patients presenting with meningoencephalitis may progress to permanent neurological deficits or ultimately die[6]. Additionally, patients may present with rare conditions, including flaccid paralysis. [7]. About 20% to 40% of Japanese encephalitis patients die during the acute stage, and 50% are left with severe neurological sequelae [8].  Patients typically have cognitive impairment, which is a major problem affecting quality of life [9]. Pigs serve as crucial amplification and reservoir hosts for JEV, with infections in pigs predominantly being subclinical [10]. The repercussions of JEV infection are notably significant in swine populations. In sows, JEV infection can lead to abortion, mummified fetuses, and stillbirth, while in boars, it can result in orchitis and a decline in semen quality, collectively causing substantial economic losses to the pig industry[11]. Although adult swine typically do not exhibit symptomatic disease following infection, JEV remains a major reproductive concern, leading to abortion, stillbirth, and congenital anomalies. Furthermore, infected piglets may present with fatal neurological disease [12]. To prevent JE, immunoprophylaxis is considered the most effective method.

      Please refer to the first paragraph of the revised manuscript for the updated Introduction section.

Comments 10: What is the life cycle of JEV?

Response 10: Thank you for the positive comments and highlighting areas for improvement.

JEV enters inside the cell through receptor mediated endocytosis. A low pH environment in the endosome triggers the viral envelope rearrangement followed by membrane fusion and genome release into cytoplasm. During the viral replication cycle, the viral genome assumes two distinct and unrelated roles. Firstly, it functions as mRNA to facilitate the translation of all viral proteins. Secondly, it serves as a template for RNA replication, thereby packaging genetic material into new virus particles. The positive sense viral RNA undergoes translation to generate a single polyprotein, which is cleaved in to structural and non-structural proteins (NSPs). Virus replicates on specialized compartments composed of ER derived membranes concentrated with viral NSPs and several host factors. The capsid protein binds to viral RNA to form nucleocapsid, which later acquired a lipid envelope consisting of the envelope and membrane proteins to form immature virion particles. The immature virus particles travel through the trans-golgi network and undergo maturation via furin cleavage[6]. The progeny virus particles are assembled via budding into the intracellular compartment and subsequently transported and released to the cell surface through the host's secretion pathway.

Comments 11: What are or could be the future treatment perspectives, knowing the role of Rab4b in JEV replication?

Response 11: Thank you for the positive comments and highlighting areas for improvement.

Understanding the role of Rab4b in JEV replication will enhance the elucidation of the interaction network between JEV and host proteins, deepen our comprehension of the JEV replication cycle, and identify potential targets for the development of novel anti-JEV therapeutics. And we added this into the revised manuscript, please see the revised manuscript for the Conclusions section in lines 1264-1267.

References for this response

  1. Ceconi, M.; Arien, K.K.; Delputte, P. Diagnosing arthropod-borne flaviviruses: non-structural protein 1 (NS1) as a biomarker. Trends Microbiol 2024, 32, 678-696, doi:10.1016/j.tim.2023.11.016.
  2. Zhang, W.; Yin, Q.; Wang, H.; Liang, G. The reemerging and outbreak of genotypes 4 and 5 of Japanese encephalitis virus. Front Cell Infect Microbiol 2023, 13, 1292693, doi:10.3389/fcimb.2023.1292693.
  3. Guodong, L. The History and Current Status of Japanede Encephalitis Pathogen Research in Chinese Mainland. Chin J Virol 2024, 40, 671-678, doi:10.13242/j.cnki.bingduxuebao.004530.
  4. Rodolfo Furlan Damiano, B.F.G., Cristiana Castanho de Rocca, Antonio de Pádua Serafim, Luiz Henrique Martins Castro, Carolina Demarchi Munhoz, Ricardo Nitrini, Geraldo Busatto Filho, Eurípedes Constantino Miguel, Giancarlo Lucchetti,Orestes Forlenza. Cognitive decline following acute viral infections: literature review and projections for post-COVID-19. European Archives of Psychiatry and Clinical Neuroscience 2022, 272, 139-154.
  5. Anup Itihas, S.J., Manish Jain, Rahul Narang, Varsha Chauhan, B. V. Tandale & Shilpa Tomar Comparison of Clinical Profile and Outcomes of Japanese Encephalitis and Acute Encephalitis Syndrome among Rural Children. Indian Journal of Pediatrics 2023, 90, 1038-1040.
  6. Sharma, K.B.; Vrati, S.; Kalia, M. Pathobiology of Japanese encephalitis virus infection. Mol Aspects Med 2021, 81, 100994, doi:10.1016/j.mam.2021.100994.
  7. Steven Grewe, M.G., Daniel B. Abrar,Torsten Feldt, Lars Wojtecki, Victor Tan, Lifea, Shazia Afzal, Sven G. Meuth, Tom Luedde,Hans Martin Orth. Myelitis with flaccid paralysis due to Japanese encephalitis: case report and review of the literature. Infection 2022, 50, 1597-1630.
  8. Weibing Shen, Y.Z., Chenguang Zhou, Yaoyao Shencorresponding. Bilateral symmetrical deep gray matter involvement and leptomeningeal enhancement in a child with MOG-IgG-associated encephalomyelitis. BMC Neurol. 2021, 21, 10, doi: 10.1186/s12883-020-02041-3. .
  9. Rong Yin, L.Y., Ying Hao, Zhiqi Yang, Tao Lu, Wanjun Jin, Meiling Dan,  Liang Peng, Yingjie Zhang, Yaxuan Wei,  Rong Li, Huiping Ma, Yuanyuan Shi, Pengcheng Fan Proteomic landscape subtype and clinical prognosis of patients with the cognitive impairment by Japanese encephalitis infection. Journal Of Neuroinflammation 2022, 19, 77, doi: doi: 10.1186/s12974-022-02439-5. .
  10. Redant, V.; Favoreel, H.W.; Dallmeier, K.; Van Campe, W.; De Regge, N. Efficient control of Japanese encephalitis virus in the central nervous system of infected pigs occurs in the absence of a pronounced inflammatory immune response. Journal of Neuroinflammation 2020, 17, doi:10.1186/s12974-020-01974-3.
  11. Nie, M.; Zhou, Y.; Li, F.; Deng, H.; Zhao, M.; Huang, Y.; Jiang, C.; Sun, X.; Xu, Z.; Zhu, L. Epidemiological investigation of swine Japanese encephalitis virus based on RT-RAA detection method. Scientific Reports 2022, 12, doi:10.1038/s41598-022-13604-4.
  12. Mansfield, K.L.; Hernández-Triana, L.M.; Banyard, A.C.; Fooks, A.R.; Johnson, N. Japanese encephalitis virus infection, diagnosis and control in domestic animals. Veterinary Microbiology 2017, 201, 85-92, doi:10.1016/j.vetmic.2017.01.014.

Round 2

Reviewer 2 Report

Comments and Suggestions for Authors

The paper is suitable for publication